# Bioengineered Ferritin Nanocarriers for Cancer Therapy

**DOI:** 10.3390/ijms22137023

**Published:** 2021-06-29

**Authors:** Xuanrong Sun, Yulu Hong, Yubei Gong, Shanshan Zheng, Dehui Xie

**Affiliations:** Collaborative Innovation Center of Yangtze River Delta Region Green Pharmaceuticals, College of Pharmaceutical Science, Zhejiang University of Technology, Hangzhou 310014, China; 2112023026@zjut.edu.cn (Y.H.); 2112023017@zjut.edu.cn (Y.G.); 2111923019@zjut.edu.cn (S.Z.); 2111923051@zjut.edu.cn (D.X.)

**Keywords:** ferritin, targeted drug delivery, surface modification, cancer therapy, phototherapy

## Abstract

Ferritin naturally exists in most organisms and can specifically recognize the transferrin 1 receptor (TfR1), which is generally highly expressed on various types of tumor cells. The pH dependent reversible assembling and disassembling property of ferritin renders it as a suitable candidate for encapsulating a variety of anticancer drugs and imaging probes. Ferritins external surface is chemically and genetically modifiable which can serve as attachment site for tumor specific targeting peptides or moieties. Moreover, the biological origin of these protein cages makes it a biocompatible nanocarrier that stabilizes and protects the enclosed particles from the external environment without provoking any toxic or immunogenic responses. Recent studies, further establish ferritin as a multifunctional nanocarrier for targeted cancer chemotherapy and phototherapy. In this review, we introduce the favorable characteristics of ferritin drug carriers, the specific targeted surface modification and a multifunctional nanocarriers combined chemotherapy with phototherapy for tumor treatment. Taken together, ferritin is a potential ideal base of engineered nanoparticles for tumor therapy and still needs to explore more on its way.

## 1. Introduction

Significant progress has been made in the diagnosis and treatment of cancer over the past few decades. While the limited efficacy of chemotherapeutics and the emergence of multidrug resistance (MDR), it is still the most common disease killer worldwide [1,2]. In the past few decades, the research of nanocarriers as drug delivery systems has received wide attention. The nanocarriers that have been developed including solid lipid nanoparticles (SLNs), liposomes, dendrimers, polymer nanoparticles (PNPs), polymer micelles, hydrogels, virus-based nanoparticles (vNPs), carbon nanotubes (CNTs), mesoporous silica nanoparticles (MSNs), gold nanoparticles, quantum dots and some other organic/inorganic hybrid nanocarriers [3,4]. Those nanocarriers can solve the key problems encountered in traditional drug therapy, such as nonspecific distribution, rapid clearance, uncontrolled drug release and low bioavailability, meanwhile, reducing toxicity and adverse reaction of drugs [5,6]. Due to the complex synthesis and drug toxicity by chemical reaction, their clinical application has been greatly restricted [7]. So far, the nanoparticle drug delivery system which approved for clinical use on present market are limited mainly including liposomal doxorubicin (DOX) and albumin-paclitaxel (Abraxane). The above nanodrug delivery system only relies on passive targeting, which can reduce toxic and side effects very well, but it does not produce significant improvement in the therapeutic index [8,9]. Under such circumstances, there is an urgent requirement to develop drug nanocarriers with high biocompatibility and tumor targeting ability.

Ferritin was first discovered in horse spleen in 1937 [10]. It is a water-soluble protein that naturally exists in most organisms [11,12]. Ferritin is a 450 kDa spherical hollow nanocage. Its outside diameter is about 12 nm and inside diameter is about 8 nm. It can bind approximately 4500 iron atoms in a nontoxic and bioavailable form [13,14,15]. The ferritin’s coating of mammals is self-assembled by 24 subunits, of which 80–90% is L-chain (light chain; 19 kDa), and 10–20% is H-chain (heavy chain; 21 kDa). There are a large number of salt bridges and hydrogen bonds between the subunits (Figure 1) [15,16]. In 2010, Seaman et al. [17] discovered that human heavy chain ferritin (HFn) can specifically recognize the highly expressed transferrin 1 receptor (TfR1) on tumor cells. This discovery greatly promoted the research of ferritin as tumor treatment nanocarriers.

According to the structure and function of ferritin, the advantages of ferritin as a drug carrier mainly including: (1) Strong tumor targeting ability: ferritin not only has the enhanced permeability and retention (EPR) effect of general nanocarriers, but also has the ability to naturally target tumor cells and be enriched at the tumor site. (2) Good biocompatibility: ferritin is naturally derived from organisms and will not produce immunogenic reactions. (3) Good stability: ferritin can withstand high temperatures up to 75 °C for 10 min and it is stable in various denaturants [15,18]. (4) Strong loading capacity of drug: Ferritin is rigid under physiological conditions, but it is still pH-dependent [19]. The structure of ferritin is decomposed into subunits in an extremely acidic (pH 2–3) or alkaline (pH 10–12) environment. When the pH returns to the physiological range, the structure of ferritin is almost completely rebuilt to contain drugs in the cage [10,19,20]. (5) Easy to modify: there are exposed lysine and cysteine residues on the surface of ferritin nanocage, which can be chemically modified to enhance tumor targeting ability [21]. All in all, ferritin is expected to become an ideal carrier for hydrophobic and non-specific cytotoxic chemotherapeutics. This review will introduce the favorable characteristics of ferritin drug carriers, the specific targeted surface modification and a multifunctional nanocarriers combined chemotherapy with phototherapy for tumor treatment.

## 2. Targeting Mechanism of Ferritin

Ferritin is proved to have the ability to naturally target tumor cells, and the initial accumulation of nanocarriers in tumors mainly depends on passive targeting. New angiogenesis and poor lymphatic drainage in tumor tissues cause the EPR effect [20,22,23]. The EPR effect is closely related to the size distribution of nanoparticles. Ferritin shows an excellent size distribution and hardly change when loading drugs, which is significantly better than chemically synthesized nanoparticles [24,25]. Due to the high iron metabolism and iron demand of tumor cells, the expression of TfR1 in cancer cells may be 100 times higher than that in normal cells [26,27]. Therefore, HFn can specifically recognize and bind to the TfR1 receptor on tumor cells. After interacting with TfR1, HFn can be effectively delivered to the lysosome through TfR1-mediated endocytosis [25] (Figure 2). It is speculated that the drugs contained in ferritin are gradually released during lysosomal acidification. In addition, there is evidence that ferritin is translocated into the nucleus by an active transport mechanism. Ferritin releases a small amount of drugs in the cytoplasm to induce a DNA damage, then the cell itself sends out harmful stimulation signals, triggering the recruitment of H-rich apoferritins into the nucleus [10]. The mechanism of how ferritin release drugs remains to be further studied.

## 3. Tumor Targeting Modification on Ferritin

Although ferritin has the ability to naturally target tumor cells, the therapeutic efficiency of a single targeted drug carrier is usually limited due to the difference in the expression of TfR1 on different tumor cells. Researchers have modified the surface of the ferritin nanocage with antibodies, peptides, etc., to give it dual-targeting characteristics, and obtain better targeting capabilities and the ability to penetrate into tumor cells (Table 1, Figure 3). At present, the methods of surface modification of ferritin mainly include modification of targeting motifs on the surface of ferritin through genetic engineering and coupling of chemical groups to lysine and cysteine residues on the surface of ferritin nanocages [28].

Peptides are currently the most commonly used tumor-targeting modification. Angiogenesis is a major feature of tumor tissues and plays an important role in the development and metastasis of tumors. There is ample evidence that the RGD polypeptide has a high affinity for the integrin αvβ3 that is highly expressed on the surface of tumor neovascularization. RGD-modified ferritin maintains its integrin selectivity and tumor targeting when loaded with various drugs (including metal ions, Dox and photosensitizers) [29]. In vivo and in vitro studies have shown that RGD-coupled ferritin nanoparticles show a stronger ability to target tumor angiogenesis than unmodified human ferritin, thereby enriching tumor tissues [30,31,32].

The tumor microenvironment has an important impact on drug delivery. Researchers have found that matrix metallo-protease (MMP) are highly specifically expressed in the tumor microenvironment. Falvo et al. [33] modified an outer masking polypeptide sequence rich in proline (P), serine (S), alanine (A) and glutamate (E) residues (PASE) on ferritin via MMP-cleavable peptide, which prolonged the half-life of the protein in the bloodstream and increased the chance of protein recycling (Figure 4). When it enters the tumor microenvironment, MMP makes the MMP-cleavable peptide fragments, so that PASE polypeptide is detached from the surface of ferritin and the ferritin activity is restored. Then, the treatment in vivo showed significant efficacy, eradicating several highly aggressive human tumors, such as pancreatic, triple-negative breast and liver cancer [34,35].

In order to enhance the targeting ability of ferritin, peptides have been developed to target receptors specifically expressed on the surface of different types of tumor cells. Epidermal growth factor receptor (EGFR) is highly expressed on the surface of breast cancer cells. Cioloboc et al. [36] conjugated an EGFR targeting peptide Ge11 to the C-terminus of the human ferritin subunit. Furthermore, SP94 peptide and ferritin can specifically bind to glucose-regulated protein GRP78 on the outside of hepatocellular carcinoma cells and accumulate at the tumor site in a hepatocellular carcinoma tumor-bearing mice model [37]. Different from general tumors, the treatment of brain tumors also considers the ability of drugs to penetrate the blood-brain barrier (BBB) and the blood-brain tumor barrier (BBTB). GKRK peptide has high affinity and specificity with heparan sulfate proteoglycan (HSPG) overexpressed in glioma cells and angiogenesis [38]. Zhai et al. [39] linked GKRK peptide to apoferritin nanocage, overcoming multiple barriers (BBB and BBTB) and obtained a precise dual targeting effect. In addition, low-density lipoprotein receptor-related protein 1 (LRP1) was found to be highly expressed in BBB and glioma cells too. Wang et al. [40] modified a new LRP1 targeting peptide on ferritin which can not only prolong the survival time of mice but also prevent brain tumors from metastasis to the spinal cord. In the above studies, intravenous injection is used, but inhalation is more effective for airway lung cancer. To avoid being blocked by mucus gel, the surface of nanoparticles can be modified with polyethylene glycol (PEG). Huang et al. [41] confirmed that in a mouse model of airway lung cancer, PEGylated FTn quickly penetrates the mucus layer and targets lung tumor tissue. Besides peptides, the folate receptor is found to be overexpressed in approximately 40% of human cancers and can mediate the endocytosis of folate conjugates. Studies have shown that folic acid, as a tumor-targeting ligand, binds to the surface of ferritin to improve the ability of ferritin to target tumors [42].

In addition, antibody modification can also be used to enhance the targeting ability of ferritin. For example, Falvo et al. [43] attached monoclonal antibodies (named EP1) to the surface of ferritin, which significantly enhanced the melanoma targeting ability of ferritin. Nanobody (Nb) is a novel antibody, which show great advantages in convenient ferritin modification compared with conventional antibodies. Liu et al. [44] realized the site-specific coupling of anti-EGFR nanobodies and ferritin nanocages, and the system can selectively accumulate in EGFR-positive A431 cancer cells (Figure 5). In addition to tumor targeted therapy, ferritin also has application value in immuno-therapy. Lee et al. [45] modified tumor-specific antigens red fluorescence protein (RFP) on the surface of ferritin, quickly targeted lymph nodes (LNs), induced a strong RFP-specific cellular immune response and successfully inhibited tumor growth in living mice. Surface modification enables ferritin to have stronger functions, and further research is needed in the future to enable ferritin to obtain better targeting ability while maintaining its stability and safety.

## 4. The Role of Ferritin in Cancer Treatment

The cage structure of ferritin gives it the potential to encapsulate a large number of drugs. In the past ten years, there have been a large number of studies on ferritin-coated chemotherapeutic drugs, and significant effects have been achieved in cell and animal experiments. In recent years, studies have found that ferritin can also be used as a carrier of photothermal agents and photosensitizers to play an important role in phototherapy, and can achieve combined treatment of chemotherapy and phototherapy (Table 2).

### 4.1. Strategies for Loading Chemotherapeutics with Ferritin

Research on ferritin as a nano-drug delivery system has a history of more than ten years. At present, there are three main drug delivery methods for ferritin (Figure 6). Most anti-cancer drugs are hydrophobic, while ferritin is water-soluble and can wrap the drugs in the cavity through hydrophobic action [55]. Moreover, the natural targeting ability of ferritin enables the effective enrichment of drugs at the tumor site. There is strong evidence that ferritin as a chemotherapeutic drug carrier can improve the efficacy of drugs, simultaneously reduce the side effects, such as cardiotoxicity of doxorubicin. The earliest successful loading of ferritin was metal nanoparticles. When ferritin is incubated with a metal ion source, since the inner cavity of ferritin has a strong negative charge, positively charged metal ions (Au, Cu, Co, etc.) can easily migrate through the pores of the ferritin nanocage and accumulate in the cavity. 

Due to the poor interaction with ferritin, the loading of non-metallic drugs in ferritin is more complicated. The ferritin shell has six hydrophobic channels and eight hydrophilic channels. With passive loading, these channels allow molecules with a maximum size of 13Å to enter, such as DOX-loaded Cu^2+^ (through hydrophilic channels) and Gefitnib (through hydrophobic channels) [29,49]. However, this method has limited drug loading capacity and does not have universal applicability. In the past ten years, drug loading methods based on dissociation have become the preferred method. The ferritin nanocage dissociates when the pH drops to 2 and reassembles after returning to neutral pH [70]. The ferritin nanocage can also be disassembled in 8M urea and reassembled after gradually reducing the urea concentration [25]. Through this method, the fat-soluble natural active compounds quercetin, curcumin, paclitaxel, etc., were successfully encapsulated in ferritin, and it was proved that the tumor treatment effect in vivo was better than that of free drugs [51,53]. However, when pH < 2.0 or at 8 M urea, HFn is completely dissociated, which often leads to the formation of soluble aggregation and insoluble precipitates, thereby impairing the recombination and recovery of HFn. In addition, acid-labile molecules have proven to be difficult to encapsulate through a pH-dependent process.

In recent years, researchers have set their sights on drug loading while maintaining the ferritin cage structure. Ferritin has a four-fold channel with an average diameter of 0.9 nm, and the channel diameter can be expanded to more than 1.0 nm through thermal fluctuations [71]. Based on the crystal structure and protein mutation analysis, Xiyun Yan group determined that there is a natural thermal-response drug entry channel on the HFn shell. Compared with the denaturation-based method, the channel-based method of loading Dox into the HFn nanocages can achieve higher drug loading efficiency, higher HFn recovery rate and better stability [72]. In addition to DOX, this new method successfully introduced more than a dozen drug molecules (metformin, nicotinamide, creatinine, famotidine, rhodamine B, etc.) into the ferritin nanocage [73]. The development of new drug loading methods is expected to promote the clinical transformation of ferritin nano-drug delivery systems.

### 4.2. Ferritin Nanoparticles Combined with Photothermal Therapy (PTT)

PTT uses photothermal agents (PTA) to absorb the near-infrared light after then efficiently convert it into heat. In cancer treatment, the high temperature generated by PTA induces apoptosis [74]. Widely used PTA are indocyanine green (ICG), semiconducting copper sulfide (CuS) nanoparticles, gold nanoparticles, carbon nanotubes and so on [75]. Due to the lack of selectivity of traditional PTT, it damages normal tissues and has not been approved for clinical use so far [76,77]. Several studies have found that using ferritin as a photothermal agent carrier can produce better photothermal efficiency while reducing toxicity. Wang et al. [57] have reported ferritin nanoparticles containing ultrasmall CuS plus laser irradiation could achieve complete tumor elimination. ICG derivatives IR820 packaged in ferritin also achieve 100% tumor elimination without obvious toxicity under laser irradiation [58]. Nevertheless, the ability of PTT to against cancer metastasis is limited due to the high resistance of cancer stem cells (CSCs) to temperature and radiation therapy [78]. Tan et al. [59] found that compared with the negative control group, ALDHhigh CSCs in the PTT group increased significantly by 2.9 times, indicating that PTT did not inhibit tumor metastasis.

To achieve better anti-cancer effects, the combination of PTT and chemotherapy has attracted researchers’ interest. Dual controlled-release nanoparticle systems with external light-heat effect and intracellular pH-triggered release were developed for combination therapy. Lin et al. [60] developed grenade-like nanoparticles with an average particle size of 60.1 nm by linking apoferritin-loaded DOX nanocages (AF-DOX NCs) with ADS-780 NIR fluorescent dye via hydrophobic interactions and electrostatic (Figure 7). After intravenous injection of grenade-like nanoparticles into mice, they passively accumulated at the tumor site due to the EPR effect. Next, the tumor site was irradiated with an 808 nm laser, and the nanoparticles dissociated to release AF-DOX NCs to actively target tumor cells and release DOX in the lysosome. In a mice HT-29 tumor model, the tumor size was significantly reduced after therapy. Another idea is to modify the ferritin inside or outside. Li et al. [61] employed Au nanoshell to modify DOX contained apoferritin. Gold nanoshell can turn light into heat due to the surface plasmon resonance properties. The highest temperature in in vivo experiments reached 59.6 °C, which can lead to necrosis of tumor cells. Moreover, a dual drug delivery system (DDDS) based on ferritin and nanoscale graphene oxide (NGO) was reported by the Guo group as efficient PTA. Firstly, NGO highly loaded resveratrol (RSV) and modified mitochondrion targeted molecule IR780 on the surface to form IR780-NGO-RSV nanoparticles (INR). Then, ferritin capsulated the INR by disassembled and reassembled. During drug delivery, DDDS released INR into the cytoplasm under the action of lysosomes. INR targeted mitochondria and released RSV to directly target organelle under NIR irradiation. In-vivo experiments demonstrate that the tumor growth was significantly inhibited, and the survival rate was high after 60 days of combined treatment [62] (Figure 8). Interestingly, Li et al. [63] discovered Prussian blue-modified ferritin nanoparticles (PB-Ft NPs) have the peroxidase-like activity which can catalyze the oxidation of chemotherapeutic drug TMB to oxidation state TMB by H_2_O_2_. It shows that PB-Ft NPs is a good tumor chemotherapy sensitizer. Therefore, it is inferred that PTT mediated by PB-Ft NPs may have the ability to reverse drug resistance. Altogether, the research on the combination of ferritin and PTA has a very broad prospect.

### 4.3. Ferritin Nanoparticles Combined with Photodynamic Therapy (PDT)

PDT has been used clinically for more than 40 years. In the presence of oxygen, the photosensitizer is activated to generate singlet oxygen or reactive oxygen species (ROS) at the tumor site under specific light irradiation [79]. ROS are strong oxidants that can directly kill tumor cells, destroy tumor vasculature to make nutrition deprived or activate the immune system to kill tumor cells [79,80]. Patients in PDT are often troubled by skin photosensitivity, and there is an urgent need to develop photosensitizers targeting tumor sites. At present, the use of ferritin as delivery vehicles of photosensitizers is in its early stages. Among them, the most widely studied is the use of hydrophilic nanoparticle ferritin as a carrier for hydrophobic photosensitizers to improve photosensitivity efficiency. Abbas et al. [64] showed that ferritin could encapsulate zinc hexadecafluorophthalocyanine (ZnF16Pc) by the interaction between zinc and internal metal-binding sites and the loading rate reached 60%. Besides, natural photosensitizer hypocrellin B (HB) had higher encapsulation efficiency in ferritin based on hydrophobic interaction, reaching 80% [65]. Because HB has the advantages of simple preparation and purification, high singlet oxygen production, and rapid in vivo metabolism, its combination with ferritin is expected to play an important role in clinical.

In addition to photosensitizers contained in the ferritin, it can also be modified on the surface of ferritin. Du et al. [66] connected the photosensitizer rose bengal (RB) to the surface of ferritin coated with DOX through an amide bond, and prepared a dual-responsive pH and ROS intelligent drug delivery system. In addition to PDT, the ROS generated by PB can also make the ferritin nanocage more prone to peptide chain breaks to release drugs. After the combined application of laser irradiation and low pH 5.0, the cumulative release rate of DOX was 92%. In in vivo experiments, tumor growth was obviously inhibited.

Cancer-associated fibroblasts (CAFs) are widely present in the periphery of a variety of tumors, acting as a physical barrier and hindering the penetration of nanoparticles into tumor tissues. Removal of CAF may reduce the content of collagen in the extra-cellular matrix, thereby improving the aggregation and diffusion of nanoparticles. Fibroblast-activation protein (FAP) is a plasma surface protein that is widely up-regulated in CAF and has been considered a universal tumor target. Li et al. [68] developed a single chain variable fragment (scFv) sequence targeting FAP conjugated and ZnF16Pc contained ferritin nanoparticles (αFAP-Z@FRT), selectively killing CAFs under the action of PDT and increasing the relative uptake rate of tumors by more than 5 times (Figure 9). Zhou et al. [69] found on the basis of the above research that PDT can induce cellular immunity to cancer cells and also stimulate immunity against CAFs. In bilateral 4T1 tumor models, after αFAP-Z@FRT were injected into animals, the primary tumors were irradiated with laser, and the contralateral tumors were unirradiated. The tumor inhibition rate of primary tumors was 90.2% on the 23rd day. At the same time, the growth of the secondary tumors was also inhibited, resulting in a non-local effect. In adoptive cell transfer study, T cells extracted from 4T1 tumor-bearing animals after αFAP-Z@FRT PDT treatment could delay the growth of A549 tumors bored in nude mice lacking native T cells. The transferred T cells should not directly kill A459 cells. The results support that αFAP-Z@FRT PDT can stimulate immunity against CAFs and induce broad-spectrum anti-cancer immunity. In this case, given that CAF exists in almost all solid tumors, anti-CAF PDT may become a unique tumor vaccination method.

### 4.4. Application of Ferritin in PTT and PDT

It is difficult to completely eliminate solid tumors using PTT or PDT alone. Multi-modal therapy combined with PTT and PDT has broad prospects in anti-multidrug resistance (MDR) and hypoxia-related tumor treatment resistance. Researchers have discovered drugs that have both PTA and photosensitizer properties. For example, after 630 nm laser irradiation, a high concentration of sinoporphyrin sodium (DVDMS) accumulates in the tumor, which not only produces reactive oxygen species (ROS), but also produces local hyperthermia. Huang et al. [30] reported that the temperature of tumor in 4T1 mice administered with DVDMS carried ferritin (FTn-DVDMS) and exposed to 630 nm laser quickly risen to 50 °C within 1 min. After 2 weeks of treatment, the tumor gradually ablated until it disappeared completely. However, laser irradiation alone or injection of FTn-DVDMS could not inhibit tumor growth. The metalla-aromatics complex named “556-Ph” has similar properties to DVDMS under laser irradiation. Zhang et al. [38] showed that the temperature of TFN@556-Ph irradiated by laser can rise by more than 50 °C, and the temperature rise remains stable after five cycles. The fluorescence intensity test confirmed that TFN@556-Ph maintains its ability to generate ROS. The tumor in the PTT + PDT group subsided, and the tumor did not recur after 16 days of treatment, proving the advantages of synergistic treatment.

## 5. Conclusions and Future Perspectives

Due to the limited curative effect of existing cancer treatment methods and severe side effects, future cancer treatments will develop prone to targeted therapy. The current developed nano drug carriers such as liposomes reduce the side effects of cytotoxic drugs in clinical trials, however without substantial improvement in long-term results. To develop effective cancer treatment, there is an urgent need for reliable nanocarriers which can target to the lesion. Many researchers have turned to study existing natural materials, especially endogenous self-assembling proteins which as new candidates for drug delivery. The therapy method using ferritin as ideal nanocarriers will become a promising way for tumor treatment according to its unique structure, natural targeting ability, good biocompatibility, biodegradability and reversible disassembly/recombination ability, as well as capability of delivering poorly soluble and non-specific cytotoxic drugs.

Due to the differences in the expression levels of TfR1 receptors on the surface of different tumor cells, the complexity of the tumor environment and limitation in vivo efficacy for some tumors. Researchers have used antibodies, peptides, antigens, etc., to modify the surface of ferritin to achieve better tumor targeting ability with the further elucidation of the pathogenesis of various tumors. Studies have shown that ferritin nanocarriers can not only improve the bioavailability of poorly soluble drugs, but also reduce the side effects of toxic drugs on normal tissues. In addition, ferritin nanocarriers significantly improve the efficacy of chemotherapeutic drugs such as DOX, cisplatin and curcumin. Furthermore, it has successfully combined ferritin with photothermal agents and/or photosensitizers to improve the efficiency of phototherapy, and realized the combined treatment of chemotherapy and phototherapy.

However, there are still some problems for further research. Firstly, ferritin cannot completely eliminate tumors due to the presence of TfR1-negative cells that cannot be specifically recognized by ferritin. Therefore, it is necessary to develop a treatment method specific to TfR1 negative cells and co-treatment with ferritin. Secondly, attaching the targeting motif to the surface of ferritin is the main strategy to improve the tumor targeting ability of ferritin nanoparticles. This strategy also brings some disadvantages, because modification of ferritin nanoparticles may increase the complexity of the industrial production and bring new issues in biosafety. On the other hand, the influence of ferritin to human body iron homeostasis needs further study. Meanwhile ferritin as a protein has great limitations and cannot be used in clinical condition at present. In addition, the preparation of ferritin nanocarriers is still at the laboratory stage, and how to achieve industrialization is a problem to be solved in the future.

Although ferritin nanocarriers as an emerging drug delivery system still has some problems to be solved, we have reason to believe that ferritin nanocarriers can be further optimized with the clarification of tumor pathogenesis and the mechanism of ferritin in vivo. In the future, the clinical application of ferritin nanocarriers is expected to play a vital role in tumor diagnosis and treatment.

## Figures and Tables

**Figure 1 ijms-22-07023-f001:**
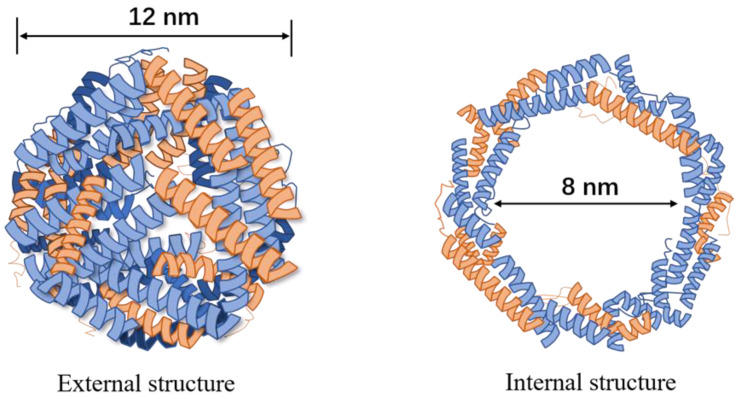
External and internal structure of ferritin.

**Figure 2 ijms-22-07023-f002:**
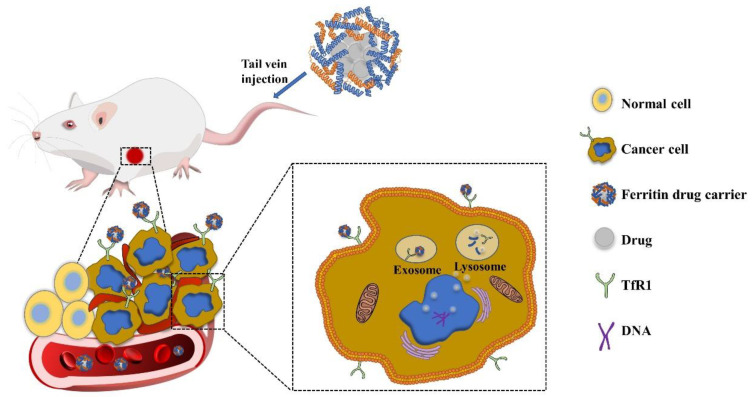
Delivery and release of ferritin drug carrier in vivo.

**Figure 3 ijms-22-07023-f003:**
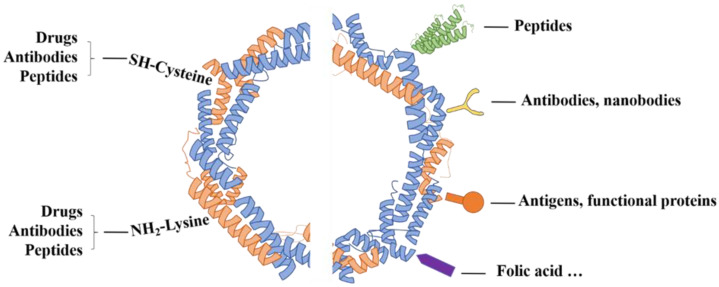
Strategies for surface functionalization of ferritin.

**Figure 4 ijms-22-07023-f004:**
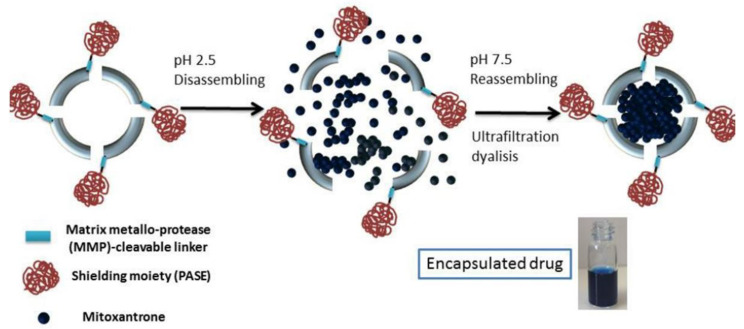
Schematic diagram of PASE polypeptide. Reproduced with permission from ref [33].

**Figure 5 ijms-22-07023-f005:**
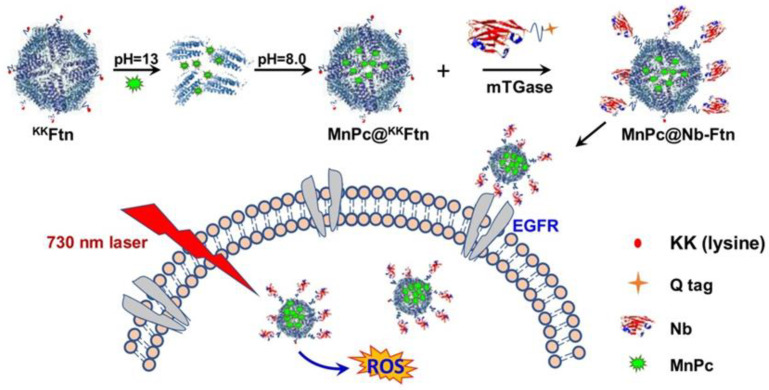
Preparation and targeted delivery of Nb-FTn. Reproduced with permission from ref [44].

**Figure 6 ijms-22-07023-f006:**
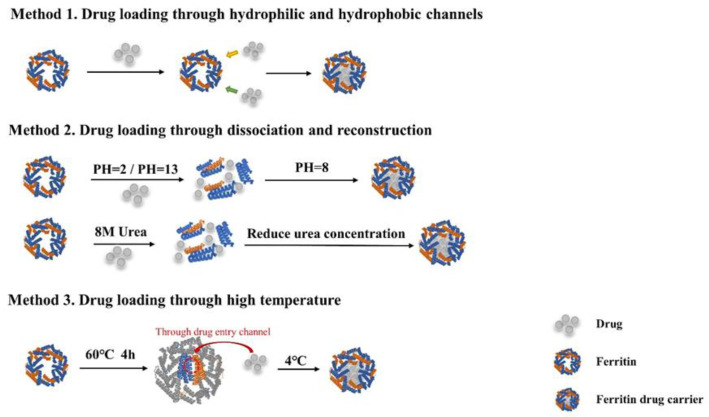
Three drug loading methods of ferritin.

**Figure 7 ijms-22-07023-f007:**
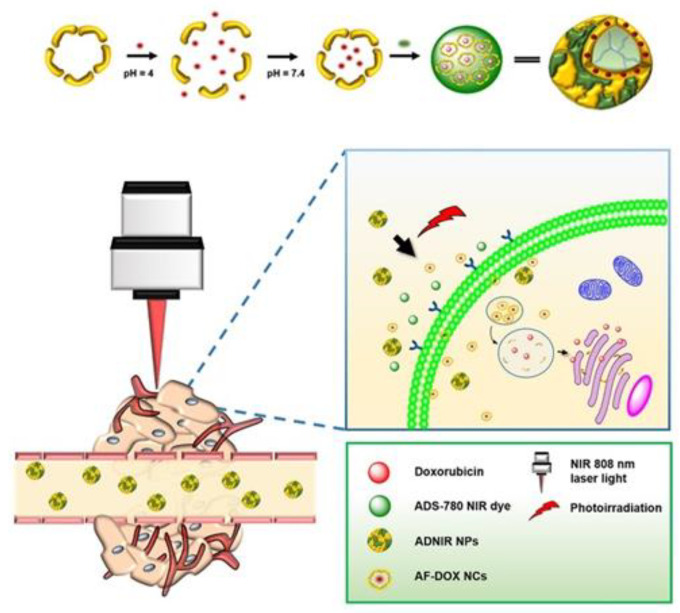
Assembly and in vivo release of grenade-like nanoparticles. Reproduced with permission from ref [60].

**Figure 8 ijms-22-07023-f008:**
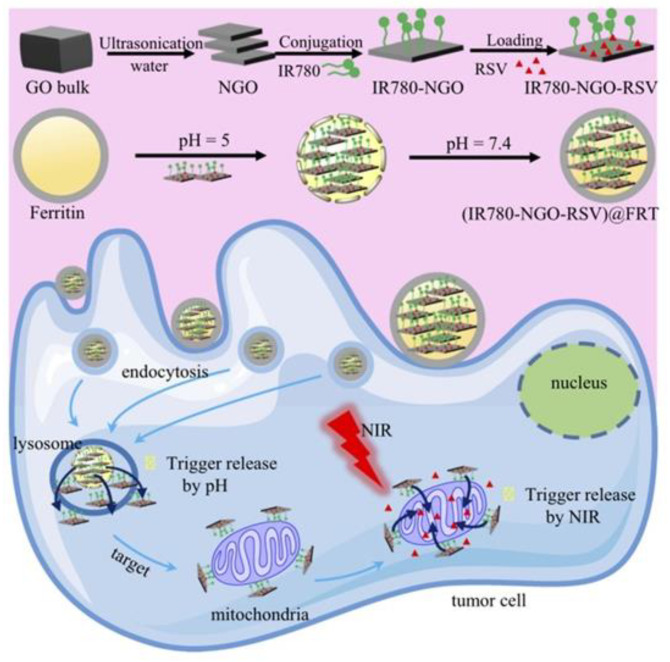
Assembly and in vivo release of drug dual-delivery system INR@FRT. Reproduced with permission from ref [62].

**Figure 9 ijms-22-07023-f009:**
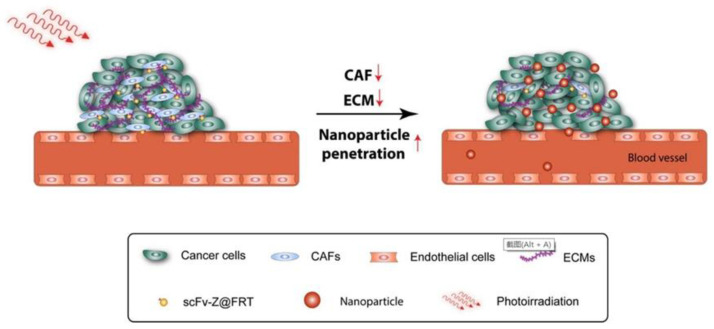
PDT affects the accumulation of ferritin nanoparticles in tumor sites. Reproduced with permission from ref [68].

**Table 1 ijms-22-07023-t001:** Surface modification of ferritin. (Please add this sentence “Table should be cited close to where it is first mentioned. Because of the reference order, please leave the table here.” for comment).

Ferritin Carrier	Surface Modification	Targeting Receptor	Cancer Type	References
Human HFn	RGD peptide	Integrin αvβ3	All types	[30]
human HFn	PASE masking peptide	\	All types	[35]
human HFn	EGF	EGFR	Breast cancer	[23]
human HFn	GE11 peptide	EGFR	Breast cancer	[36]
*Pyrococcus furiosus* ferritin	SP94 peptide	GRP78	Liver cancer	[37]
Apoferritin	GKRK peptide	HSPG	Glioma	[39]
Human HFn	Angiopep-2 peptide	LRP1	Glioma	[40]
Human HFn	PEG	\	Airway lung cancer	[41]
Human HFn	folic acid	folic acid receptor	\	[42]
Human HFn	Ep1 antibody	CSPG4	Melanoma	[43]
Human HFn	Nb	EGFR	Epidermoid carcinoma	[44]
Human HFn	RFP	LNs	Melanoma	[45]
Human HFn	MSH	Melanoma cells	Melanoma	[46]

Abbreviations: EGF: epidermal growth factor; CSPG4: chondroitin sulphate proteoglycan 4; RFP: red fluorescence protein; LNs: lymph nodes; MSH: melanoma targeting peptide.

**Table 2 ijms-22-07023-t002:** Drug loading of ferritin.

Drug Carrier	Encapsulated Drug	Application	Cancer Type	Treatment Effect	References
Human HFn	Cisplatin	Chemotherapy	Melanoma	Improved the therapeutic index of melanoma	[43]
Horse spleen ferritin	Gold-based anticancer drugs	Chemotherapy	All types	The impact on normal cells was significantly reduced	[47,48]
Human HFn (MSH and PEG modified	Co (II)	Hyperthermia	Melanoma	Cell viability significantly reduced	[46]
Human HFn	Gefitinib	Chemotherapy	Breast cancer	Has enhanced tumor suppression (GI_50_ = 0.52 × 10^−6^ M) compared to free Gefitinib (GI_50_ = 1.66 × 10^−6^ M)	[49]
Human HFn	DOX (Pre-complexation with Cu (II))	Chemotherapy	Glioblastoma	89.6% TGI of U87MG subcutaneous tumor models	[29]
Human HFn	DOX	Chemotherapy	Gastric cancer	91.1% TGI of TfR1-positive gastric cancer models	[50]
Apoferritin	VCR	Chemotherapy	Glioma	Relative tumor proliferation rate of VCR-loaded apoferritin (36.31 ± 5.52%) was much lower than free VCR (96.34 ± 5.56%)	[39]
Human HFn	Paclitaxel	Chemotherapy	Breast cancer	The tumor volume in Taxol group (0.8 cm^3^) was much lower than PBS group (2.26 cm^3^)	[51]
Apoferritin	Curcumin	Chemotherapy	Breast cancer	The therapeutic dose reached 97 μg/mL	[52]
Apoferritin	Quercetin and curcumin	Chemotherapy	Breast cancer	The EC_50_ for MCF7 reduced to 11 μM	[53]
Human HFn	Atropine	Chemotherapy	Pancreatic cancer	The neurogenesis in pancreatic cancer was impaired	[54]
Human HFn (PASE masking peptide modified)	Genz-644282	Chemotherapy	All types	94.0% TGI of xenograft (subcutaneous) model of pancreatic (HPAF II cells) cancer model; 100% TGI of xenograft PaCa44 pancreatic, triple-negative breast and liver cancer model	[34,35]
Apoferritin	GW 610 and amino acid prodrugs	Chemotherapy	Breast and colorectal carcinoma	The Apoferritin-encapsulated Lys modified GW 608 complexes exhibit potent anticancer activity	[55]
Horse spleen ferritin (protective PAS peptides or PEG modified)	Ellipticine	Chemotherapy	Breast cancer	All three surface modifications of ferritin displayed beneficial effects on biocompatibility	[56]
Human HFn	CuS	PTT	Glioblastoma	100% tumor elimination was achieved in CuS-Fn group plus laser irradiation	[57]
Human HFn	IR820	PTT	Breast cancer	Eliminated 100% mouse breast cancer cells	[58]
Apoferritin	Epirubicin and DBN	Chemotherapy and PTT	Breast cancer	Killed about 80% of CSCs in primary tumor with photodynamic therapy	[59]
Apoferritin	DOX and ADNIR	Chemotherapy and PTT	Colon cancer	The tumor size was significantly reduced in a mice HT-29 tumor model	[60]
Apoferritin (with Au nanoshell)	DOX	Chemotherapy and PTT	Liver cancer	Hepa1-6 cells have a low viability (4.3%) after chemotherapy and PTT	[61]
Apoferritin	RSV and IR780	PTT	Ovarian cancer	The survival rate was high after 60 days of combined treatment	[62]
Human HFn (Prussian blue PB-modified)	gemcitabine GEM	Chemotherapy and PTT	Breast cancer	PB-Ft NPs-assisted photothermo-chemotherapy effectively damaged the 4T1 tumor cells	[63]
Apoferritin (RGD modified)	ZnF16Pc	PDT	Breast cancer	The loading rate reached 60%	[64]
Apoferritin	HB	PDT	Breast cancer	HB encapsulation efficiency of 85%	[65]
Apoferritin	DOX and RB	Chemotherapy and PDT	Breast cancer	The cell inhibition rate was up to ~83%	[66]
Human HFn (CDs modified)	DOX	Chemotherapy and PDT	Breast cancer	Simultaneous action of CDs and DOX was the most effective for DNA damage	[67]
Human HFn (scFv sequence modified)	ZnF16Pc	PDT	Breast cancer	Selectively killing CAFs under the action of PDT; stimulate immunity against CAFs and induce broad-spectrum anti-cancer immunity	[68,69]
Human HFn (RGD peptide modified)	DVDMS	PTT and PDT	Breast cancer	Eliminated 100% mouse breast cancer cells with PTT and PDT	[30]
Human HFn (CGKRK peptide modified)	556-Ph	PTT and PDT	Breast cancer	The tumor in the PTT + PDT group did not recur after 16 days of treatment	[38]

Abbreviations: TGI: tumor growth inhibition; VCR: vincristine sulfate; GW 610: 5-fluoro-2-(3,4-dimethoxyphenyl) benzthiazole; GW 608: 5-fluoro-2-(4-hydroxy-3-methox phenyl) benzothiazole; CuS: ultrasmall copper sulfide; IR820: new cyanine green; DBN: 1,1-dioctadecyl-3,3,3,3-tetramethylindotri-carbocyanine iodide; ADNIR: ADS-780 near infrared fluorescent dye.

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
