# Peer review of "Bioengineered Ferritin Nanocarriers for Cancer Therapy"

_ijms, 2021, doi:10.3390/ijms22137023_

Round 1

Reviewer 1 Report

The present manuscript reviews the recent advances in the use of ferritin nanocarriers for cancer therapy, investigating the targeting mechanism, modifications and role in cancer treatment. Compared to the old version, the authors made a great effort, revolutionizing the entire manuscript. In my opinion this work is well written, is readable and offers a clear view of the state of the art in this field.

I would like to suggest to introduce in table 2 also the relative cancer type, as done in table 1, combining the informations previously displayed in table 1 of the old version.

Author Response

Response to Reviewer 1 Comments

Point 1: The present manuscript reviews the recent advances in the use of ferritin nanocarriers for cancer therapy, investigating the targeting mechanism, modifications and role in cancer treatment. Compared to the old version, the authors made a great effort, revolutionizing the entire manuscript. In my opinion this work is well written, is readable and offers a clear view of the state of the art in this field.

Response 1: Thank you very much for your positive comments.

Point 2: I would like to suggest to introduce in table 2 also the relative cancer type, as done in table 1, combining the informations previously displayed in table 1 of the old version.

Response 2: Thank you for your kind suggestions. We have supplemented the relative cancer type as suggested in table 2 (page 11, line 429 to page 13, line 456).

Explanation of additional changes

Point 1: We changed the first letter of the 3 words in table 1 to uppercase (Ref. 31, Ref.36, Ref.42).

Point 2: Reference 30 in Table 2. In the old version, the references and content correspond to the wrong ones and have been changed.

Point 3: Beautify the schematic diagram of cancer cells in figure 2, supplementing cell membranes, endoplasmic reticulum, and mitochondria.

Point 4: Change figure 4 to the schematic in Ref. 34.

Reviewer 2 Report

The manuscript by Xuanrong et al. is a comprehensive review on the ferritin-based nanocarriers as drug delivery systems. In particular, the Authors aim is to describe the favorable characteristics of ferritin drug carriers, the specific targeted surface modification and the use of these systems as multifunctional nanocarriers combining chemotherapy with phototherapy for tumor treatment. Overall, these aspects are effectively reported and discussed in detail inside the review, but some changes are needed, especially in the references section. Below my suggestions to improve the manuscript:

- Page 3, line 88. Although a mechanism on how ferritins release drugs inside the cell is not fully elucidated, some hypotheses are discussed and reported. Please include these papers: 1) Bellini et al., Journal of Controlled Release 196 (2014) 184–196.

- Page 4, lanes 117-125. Please define better the PASE peptide. I suggest:” an outer masking polypeptide sequence rich in proline (P), serine (S), alanine (A) and glutamate (E) residues (PASE).

In addition, the refs 33 and 34 should be reversed because the ref 34 is the corrected one for the first PASE-based ferritin variant.

- Figure 6 and throughout the text. The Authors use the sentence:” drug DELIVERY methods” to describe the drug LOADING methods. Please change “delivery” with “loading” in the figure legend and in the text (lanes 211-234)

- Line 212, “denaturation” should be changed in “dissociation”

- Table 2. The application for Ref 46 should be Hyperthermia instead of Chemotherapy. Please correct. In ref. 33, a 100% TGI was obtained for the xenograft PaCa44 pancreatic cancer. Please also include this information. Moreover, the molecule Genz-644282 was reported also in this ref: Falvo et al., Pharmaceutics. 2020 Oct 20;12(10):992. doi: 10.3390/pharmaceutics12100992. Please include it together with ref. 33 (to be changed in 34… see above)

In addition, some references should be added here: 1) Ref. 47; 2) Tesarova et al., APPLIED MATERIALS TODAY , p. 1 - 11, doi: 10.1016/j.apmt.2019.100501

Author Response

Response to Reviewer 2 Comments

Point 1: The manuscript by Xuanrong et al. is a comprehensive review on the ferritin-based nanocarriers as drug delivery systems. In particular, the Authors aim is to describe the favorable characteristics of ferritin drug carriers, the specific targeted surface modification and the use of these systems as multifunctional nanocarriers combining chemotherapy with phototherapy for tumor treatment. Overall, these aspects are effectively reported and discussed in detail inside the review, but some changes are needed, especially in the references section.

Response 1: Thank you very much for your positive comments.

Point 2: Page 3, line 88. Although a mechanism on how ferritins release drugs inside the cell is not fully elucidated, some hypotheses are discussed and reported. Please include these papers: 1) Bellini et al., Journal of Controlled Release 196 (2014) 184–196.

Response 2: We thank the reviewer for the suggestion. We have supplemented the mechanism of ferritins releasing drugs in the nucleus mentioned in Ref. 28 (page 3, lines 87-91). As mentioned in this article, ferritin releases a small amount of drugs in the cytoplasm to induce a DNA damage, then the cell itself sends out harmful stimulation signals, triggering the recruitment of H-rich apoferritins into the nucleus.

Point 3: Page 4, lanes 117-125. Please define better the PASE peptide. I suggest:” an outer masking polypeptide sequence rich in proline (P), serine (S), alanine (A) and glutamate (E) residues (PASE).

In addition, the refs 33 and 34 should be reversed because the ref 34 is the corrected one for the first PASE-based ferritin variant.

Response 3: We thank the reviewer for the suggestions. We have supplemented the definition of PASE peptide according to your suggestion (page 4, lines 128-129). In addition, we have changed the order of references using PASE peptides to shield ferritin. Ref. 35 has been added, and ref. 35 in the previous edition has been deleted to better explain the research of Falvo et al. on ferritin modified by PASE polypeptide (page 4, lines 128-136). Moreover,

Point 4: Figure 6 and throughout the text. The Authors use the sentence:” drug DELIVERY methods” to describe the drug LOADING methods. Please change “delivery” with “loading” in the figure legend and in the text (lanes 211-234)

Response 4: We are so sorry to make a mistake. Thanks to the reviewer for pointing out our inappropriate words. We have changed the corresponding "delivery" in the text to "loading" (page 7 line 246, page 7 line 268, page 8 line 284).

Point 5: Line 212, “denaturation” should be changed in “dissociation”

Response 5: We are so sorry to make a mistake. Thanks to the reviewer for pointing out our inappropriate words, we have changed it in page 7 line 247.

Point 6: Table 2. The application for Ref 46 should be Hyperthermia instead of Chemotherapy. Please correct. In ref. 33, a 100% TGI was obtained for the xenograft PaCa44 pancreatic cancer. Please also include this information. Moreover, the molecule Genz-644282 was reported also in this ref: Falvo et al., Pharmaceutics. 2020 Oct 20;12(10):992. doi: 10.3390/pharmaceutics12100992. Please include it together with ref. 33 (to be changed in 34… see above)

In addition, some references should be added here: 1) Ref. 47; 2) Tesarova et al., APPLIED MATERIALS TODAY , p. 1 - 11, doi: 10.1016/j.apmt.2019.100501

Response 6: We thank the reviewer for the suggestions. We are so sorry to make mistakes in table 2 and have replace Hyperthermia instead of Chemotherapy (Ref. 47). In addition, add Ref. 35 with Ref. 36 and corrected treatment effect information. Moreover, added Ref. 48 and Ref. 81 to table 2.

Explanation of additional changes

Point 1: We changed the first letter of the 3 words in table 1 to uppercase (Ref. 31, Ref.36, Ref.42).

Point 2: Reference 30 in Table 2. In the old version, the references and content correspond to the wrong ones and have been changed.

Point 3: Beautify the schematic diagram of cancer cells in figure 2, supplementing cell membranes, endoplasmic reticulum, and mitochondria.

Point 4: Change figure 4 to the schematic in Ref. 34.
